# Predicting species assemblages at wildlife crossing structures using multivariate regression of principal coordinates

Thomas J. Yamashita[1¤a,¤b]*, Daniel G. Scognamillo[1¤c], Kevin W. Ryer[2], Richard J. Kline[2], Michael E. Tewes[1], John H. Young Jr[3], Jason V. Lombardi[1¤d]

1 Caesar Kleberg Wildlife Research Institute, Texas A&M University – Kingsville, Kingsville, Texas, United States of America, 2 School of Earth, Environmental, and Marine Sciences, University of Texas Rio Grande Valley, Port Isabel, Texas, United States of America, 3 Environmental Affairs Division, Texas Department of Transportation, Austin, Texas, United States of America

¤a Current address: Rocky Mountain Research Station, United States Forest Service, Fort Collins, CO 80526, USA
¤b Current address: Department of Fish, Wildlife, and Conservation Biology, Colorado State University, Fort Collins, CO 80523, USA
¤c Current address: Safari Club International Foundation, San Antonio, TX 78245, USA
¤d Current address: Wildlife Health Laboratory, California Department of Fish and Wildlife, Rancho Cordova, CA 95670, USA
* tjyamashta@gmail.com

## Abstract

Wildlife populations are in decline due to human threats, including highways. Strategies for reducing road impacts on wildlife include wildlife fencing which keep animals off roads and wildlife crossing structures (WCSs) which provide safe passage across roads. Wildlife crossing structures are diverse and transportation managers are often interested in identifying which WCS designs are effective for target species so a model that predicts target species usage of WCSs is likely to be beneficial to managers and biologists. Wildlife crossing structures are typically built for select species but are utilized by other species, so it may be beneficial to examine WCS use at the community level. We used camera trap data to develop a predictive model of mammal community composition at WCSs built for ocelots (*Leopardus pardalis*) to predict total detections, successful crossings, and failed crossings using spatial, temporal, structural, environmental, and anthropogenic characteristics. During the first-year after construction of WCSs, structural and anthropogenic characteristics of the WCSs were more important than the environmental characteristics although we expect environmental characteristics to become more important with time. Our models reasonably predicted total detections but were less effective at predicting successful and failed crossings, likely due to potential finer-scale, more dynamic effects like noise or microclimate conditions that may drive an animal's decision to use a WCS. While our study focused on WCSs built for ocelots, to our knowledge, our model is the first model of WCS effectiveness for mammal communities and provide a generalized

**Data availability statement:** Code and other data used in this manuscript are available on FigShare (DOI: 10.6084/m9.figshare.30117358). Due to the presence of federally endangered species in the camera trap data, camera trap data can be made available upon request. Please contact Dr David Hewitt, Director of the Caesar Kleberg Wildlife Research Institute, at David.Hewitt@tamuk.edu to request camera data. The functions used for processing camera data are available in the cameraTrapping package on GitHub at https://github.com/tomyamashita/cameraTrapping (DOI: 10.5281/zenodo.17081060).

**Funding:** This work was funded by the Texas Department of Transportation.

**Competing interests:** The authors have declared that no competing interests exist.

framework for predicting WCS use which can be applied anywhere where WCSs are being built.

## Introduction

Globally, wildlife populations are in decline due, in large part, to human activity [1,2]. A major source of these declines stems from the construction and maintenance of roads through natural habitats [3,4]. Roads and associated vehicle traffic create barriers to connectivity by preventing animals from successfully crossing the road, either through vehicle collisions or avoidance behaviors [5]. This loss of connectivity leads to fragmentation effects and associated population declines [6]. Roads, however, are critical infrastructure for people, and new construction and expansion of roads are often essential components of development and growth [7]. Therefore, development of effective road mitigation structures for wildlife is critical to balancing human needs with wildlife conservation [4].

While many different strategies exist for reducing road impacts on wildlife, they tend to group into two methods: (1) keeping animals off roads and (2) allowing animals to cross roads safely [3]. For example, fencing keeps animals off roads while wildlife crossing structures (WCSs) allow animals to cross roads. These two techniques are often used in conjunction because fencing alone increases fragmentation [8] while WCSs are often less effective without fencing [9].

While the factors that make an effective fence are fairly straightforward, the factors that make an effective WCS are more complex. The definition of a WCS is simply a structure built under or over a road that provides safe passage for wildlife across said road. Construction of WCSs is typically targeted towards particular species such as large migratory ungulates, amphibians, or endangered species [10–12]. Therefore, WCS designs are often species-specific; however, general trends have been established for some mammalian functional groups. For example, deer (*Odocoileus spp.*) and other ungulates prefer overpasses or wide, open bridge-style underpasses [13–15]; while many carnivores prefer smaller, darker culvert-style underpasses [16,17]. Whereas historically WCSs were built for migratory species, currently they are also being built for elusive carnivores such as mountain lions (*Puma concolor* [18,19]), tigers (*Panthera tigris* [20]), and ocelots (*Leopardus pardalis* [11,21]). Therefore, understanding how different species use WCSs, particularly non-migratory species, will help improve efforts to provide functional and effective road mitigation structures for wildlife.

An important aspect of understanding WCS use by non-migratory species is to identify the WCS characteristics that target species prefer. Many factors affect WCS use, including spatial and temporal characteristics (to account for autocorrelation), structural characteristics of the WCS such as the dimensions of the WCS and its openness, substrate, and the length of fencing around the WCS, environmental characteristics including land cover around the WCS or the presence of water features, and anthropogenic characteristics such as human and associated livestock or domestic animal activity at the WCS and vehicle traffic and speed [4, Ch. 15, 22].

To add another layer of complexity, responses towards particular characteristics have been shown to be species-specific [16], yet preferences for particular WCS characteristics effects may manifest at the community level as well. Additionally, dispersing target individuals may only be expected to use a WCS once every few years so assessing effectiveness for those species may not be feasible over the duration of WCS monitoring. Finally, greater diversity typically includes more rare species so a diversity-based approach may allow researchers to determine the likelihood that a target species will use a WCS even if it is not detected during the study period. Therefore, examining community composition (a measure of beta diversity [23]) at WCSs may provide useful information about WCS use by target species.

Typically, early WCS designs are based on assumptions about target species preferences, then combined with engineering constraints to generate a WCS that hopefully works. Whereas these designs are not always effective, using information from previous WCS designs allows transportation planners to design functional WCSs for target species. Therefore, models demonstrating the effectiveness of early WCSs can be used in future WCS designs.

The objective of this study was to assess the effectiveness of WCSs for the terrestrial medium-large mammal community (hereafter mammal community) in South Texas over the first year of WCS post-construction and to develop a predictive model to estimate WCS use at future WCSs. We employed a community composition approach to examine how WCS characteristics impact the mammal community at WCSs and ultimately inform conservation of ocelots, an endangered species in the region. We expected diversity at WCSs to increase over time as animals learned about the structures and would be most associated with WCS structural characteristics [16]. Additionally, sites with ocelots present were expected to be the most diverse and models based on structural, environmental, and anthropogenic characteristics of community composition were likely to have higher predictive power to estimate ocelot use at future WCS locations in the region.

## Methods

### Study area

Our study area was the Lower Rio Grande Valley (LRGV) ecological region of South Texas. The region has experienced significant native habitat loss and anthropogenic growth, resulting in increased urban development, rangeland conversion, and road network expansion [24]. The LRGV has high biodiversity and several rare and endangered species, such as the white-nosed coatimundi (*Nasua narica*) and aplomado falcon (*Falco femoralis* [25]), are native to the region. Native vegetation of the LRGV includes a mix of Gulf Coast cord grass (*Spartina spartinae*) prairie, salt flats, and thornscrub (a diverse assemblage of short, thorny, and dense trees and shrubs [26–28]). The region experiences hot summers and mild winters and periodic rainfall [29].

Anthropogenic expansion has threatened many species, especially ocelots. Ocelots are a medium-sized felid, native to the Americas, ranging from Uruguay and southern Brazil to northern Mexico, with isolated populations in southern Texas, USA [30]. In the USA, ocelots are endangered, with only two known small populations occurring on working private ranchlands (Ranch Population; > 60 individuals) and around Laguna Atascosa National Wildlife Refuge (Refuge Population; < 16–20 individuals [31,32]) with the Refuge population being more threatened by roads and urban expansion [24,32].

Currently, the largest source of known mortality for ocelots is vehicle collisions [33,34]. To address these challenges, the Texas Department of Transportation (TxDOT), in partnership with the US Fish and Wildlife Service (USFWS), built WCSs along highways near the Refuge ocelot population to reduce the threat of roads to ocelots, including both resident [35] and dispersing individuals [11]. Ocelots are associated with dense vegetative cover in South Texas, so WCS construction has focused on past ocelot-vehicle collision sites and vegetation corridors that could facilitate dispersal across highways [11]. However, vehicle collision sites may not represent appropriate WCS locations in part because vegetation structure near successful road crossing events are different than roadkill locations [36] and because current roadkill locations may not represent historic populations or movement patterns [37].

Because little was known about ocelot WCS use, a wide variety of WCSs were designed, including bridge-style and culvert-style underpasses, various types of concrete steps, collectively known as catwalks, and different lengths of fencing around the WCS. Eighteen WCSs on three different highways in Cameron County were constructed between 2013

and 2022 with more planned over the next 10 years [11]. Therefore, it is necessary to identify effective WCS designs for ocelots. However, due to low local population sizes, ocelots rarely use WCSs, so a community composition approach may provide a better assessment of potential use of WCSs by ocelots.

### Wildlife crossing structure monitoring

In Cameron County, 18 WCSs have been constructed on three highways located near the Laguna Atascosa National Wildlife Refuge: five WCSs on State Highway (SH) 100, eight WCSs on Farm-to-Market (FM) 106, and five WCSs on FM 1847 (Fig 1). The three highways have experienced multiple ocelot-vehicle collisions over the last 40 years and thus were selected as optimal locations for WCSs. State Highway 100 is a four-lane highway with a concrete median, while FM 106 and FM 1847 are two-lane highways with no median barrier (Table 1). All three highways have speed limits between 88 and 105 km/h and the annual average daily traffic (AADT) varied among the highways from 743 on FM 106–7778 on SH 100 (Table 1 [38]).

Construction of WCSs on SH 100 began in 2016 and ended in May 2018 (Table 2). One WCS was preexisting and only received minor modifications but remained open during construction. The first new WCSs opened to animals in October 2017. Construction on FM 106 began in 2015 and ended in December 2019; WCSs opened to animals in July 2019. On

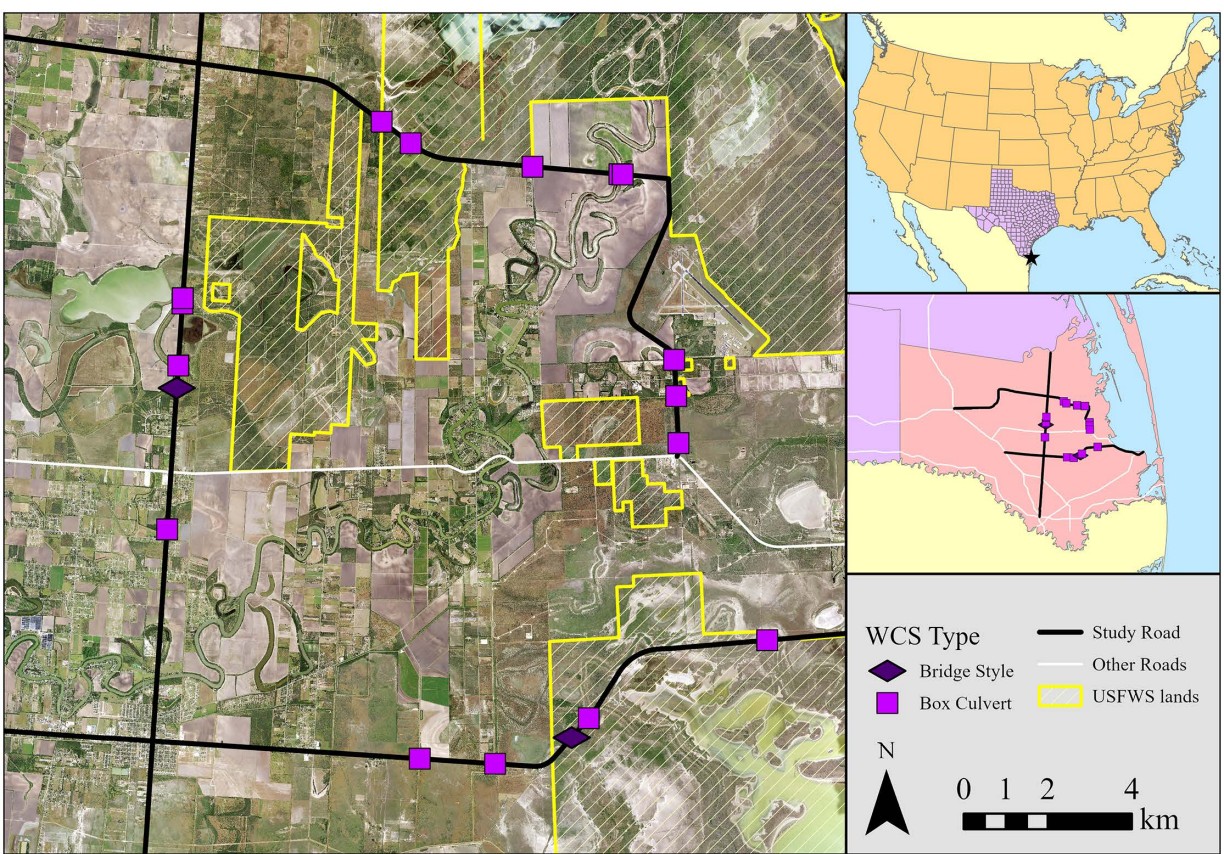

**Fig 1. Study Area: Study area showing the locations of 18 wildlife crossing structures (WCS) constructed on three highways in Cameron County, Texas, USA between 2016 and 2022.** The basemap is freely available from the National Agriculture Imagery Program, US Department of Agriculture.

FM 1847, construction began in 2020 and ended in August 2022, with the first WCSs opening to animals in January 2022 (Table 2).

The WCSs were either box culverts or bridge-style underpasses (Fig 1; S1 Appendix), which aid in flood prevention during heavy rainfall events. In addition, WCSs at the greatest risk of flooding received catwalks, made of either concrete or dirt to facilitate animal use during flooding. The WCSs varied in size and substrate: concrete box culverts, box culverts with permanent or intermittent water, and bridges with dirt substrates (S1 Appendix).

Camera trap monitoring began as soon as possible after the WCS opened. Two to 12 Reconyx Hyperfire PC900 or Reconyx Hyperfire2 (Reconyx Inc., Holmen, WI, USA) camera traps were deployed at each site to record animal use in and around the WCS. Once construction activities concluded at a WCS, we added an active infrared trigger (AIR) system to aid in determining WCS use. The AIR was an infrared beam across the entrance of a WCS that triggers a camera when it is broken and has been shown to increase detections of animals at WCSs in South Texas [39]. Camera trap sites were checked every two to four weeks to exchange memory cards, perform camera maintenance, and clear vegetation at the WCSs to accurately capture animal use of WCSs. Access to WCSs was provided by the Texas Department of Transportation and no permits were required because we had no contact with animals.

Once memory cards were collected, we downloaded the data and sorted photographs using one of two techniques (Table 1). All terrestrial mammals except rats and mice (collectively called rodents) were classified to species (S2 Appendix). If an individual could not be identified to the species level, it was identified to the lowest taxonomic rank (e.g., unknown mammal or unknown ungulate). Only these species were examined because they represent terrestrial mammals that are likely to be affected by roads. Photographs from SH 100 and FM 106 were sorted manually and placed into folders representing the species and number of individuals in the photograph [40,41]. For data collected on FM 1847, the

**Table 1. Characteristics of the study roads showing the road characteristics, number of wildlife crossing structures (WCS), and camera monitoring and sorting protocols.**

|  | FM106 | FM1847 | SH100 |
|---|---|---|---|
| Number of WCS | 8 | 5 | 5 |
| Vehicle traffic[1] | 743–1742 | 2050–2793 | 7048–7778 |
| Fencing length[2] | Moderate | Short | Long |
| Urbanization | Rural | Urban – Rural | Rural |
| Cameras per site | 3–4 | 8 | 3–10 |
| Camera trap nights | 550 for all sites | 546–596 | 526–982 |
| Megadetector AI | No | Yes | No |
| Sorting Protocol | "manual" | Timelapse | "manual" |

[1]Vehicle traffic is the average annual daily traffic and is measured in vehicles per day.

[2]Fencing length is categorized here based on the average length of fencing along all WCSs on each highway. For individual fencing lengths, see S1 Appendix.

**Table 2. Timeline of wildlife crossing structure (WCS) construction showing the starting year of construction, the opening date for WCSs, the end date of construction, and the end of monitoring at the WCSs for this study.**

|  | FM106 | FM1847 | SH100 |
|---|---|---|---|
| Construction Start | 2013 | 2020 | 2016 |
| WCS Open Date[1] | July 2019 | January 2022 | September 2016 |
| Construction End | December 2019 | August 2022 | May 2018 |
| End Monitoring | December 2020 | August 2023 | May 2019 |

[1]WCS opening date is the month the first WCS on a particular highway opened to animals. For individual WCS opening dates, see S1 Appendix.

Microsoft MegaDetector artificial intelligence program aided in our identification of animals, people, vehicles, and empty images (false captures [42]). Using previous data from FM 1847, confidence thresholds of 0.55 for animal, person, or vehicle and 0.60 for empty images were identified that balance the true positive and false negative rates (TJY, unpublished data). Photographs identified as a false capture but not an animal, person, or vehicle (i.e., above the 0.60 threshold for empty images but below the 0.55 threshold for the other categories) were labelled as false captures and not examined. Photographs were sorted to the lowest taxonomic level and the number of individuals counted using the program Timelapse2 [43]. Photographs were then placed into the same folder structure as the "manual" method used on SH 100 and FM 106 using Program *R* [44].

Once the camera data were sorted, we assessed how individuals interacted with WCSs using the system developed by Kintsch and Cramer [45] and modified for use in South Texas by Cogan [39]. Photographs were grouped by WCS with an independent event considered to be a 30-minute interval between photographs of the same species on any camera at a WCS [46]. We classified each independent event into one of five different interactions: (A) successful crossing where an animal passes from one side to another through a WCS, (B) entry/exit where an animal enters a WCS and exits on the same side, (C) approach where an animal approaches a WCS but does not enter, (D) detection where an animal is detected at a WCS but does not interact with it, or (E) day bed which is a special type of "B" interaction where an animal enters a WCS to rest or forage and does not exit for several hours. Interactions on FM 106 were not assessed between July 2019 and December 2019 (the construction period) so number of events was calculated using a 30-minute interval at each WCS. For "A" interactions, animals sometimes crossed back and forth during the same independent event. In these cases, we assigned two "A" interactions, one in each direction, but did not consider any additional crossings during that event. When multiple individuals were detected during a single event but had different interactions, we assigned individuals to their respective interaction type.

We then grouped these interactions into three categories for analysis: total detections (the number of interaction events at each WCS), the number of successful crossings (the number of "A" interactions), and the number of failed crossings ("B" + "C" + "E" interactions). Construction-period events on FM 106 were only included in the total detections analysis. Total detections, successful crossings, and failed crossings of each species were summarized by month and normalized to the number of days in the month to calculate the number of interactions per day for each species in a month.

While monitoring of WCSs will continue for three to five years after construction of the WCSs, we limited our analyses to the time the WCS opened until the end of the first year after construction was completed. Predictors of WCS use change over time [14,47] so we limited our study to the period when animals were learning about the WCSs.

## Collection and processing of predictor variables

We identified a suite of 14 potential predictors representing structural, environmental, and anthropogenic characteristics of WCSs with coordinates in UTM format and month to account for spatial and temporal autocorrelation (Table 3). We detrended spatial and temporal autocorrelation using distance-based Moran's eigenvector maps (dbMEMs [51,52, Ch. 14]). A dbMEM is a generalized principal coordinates of neighbor matrices (PCNM) analysis and is a principal coordinates analysis (PCO) on a truncated pairwise geographic distance matrix where the PCO axes represent the autocorrelation at different spatial or temporal scales [53]. A dbMEM maximizes the Moran's I autocorrelation coefficient between two sites and generates a set of orthogonal axes that can be used as predictors in community composition models [54]. We calculated dbMEMs using the pcnm function in the *vegan* package in R [55]. We determined the threshold distance using the spantree function in *vegan* which represented the longest distance that connects two sites or times [52,54]. We examined plots of each dbMEM axis against the original variables (either the coordinates or month) to identify the scales at which each axis operates (S3 Appendix [53]). We generated separate dbMEMs for the total detections analysis and the successful/failed crossings analyses due to differences in sample size between the analyses.

**Table 3. Characteristics of the predictor variables showing the data source, predictor data set (Group; spatial = spat, temporal = temp, structural = str, environmental = env, or anthropogenic = ant), data type (Type; numeric = num or categorical = cat), mean ± standard deviation (SD).**

| Predictor | Source[1] | Group | Type | Mean ± SD | Units |
|---|---|---|---|---|---|
| UTM X | This study | spat | Num | — | Meters |
| UTM Y | This study | spat | Num | — | Meters |
| Construction month | This study | temp | Num | — | Month |
| Openness ratio | This study | str | Num | 0.338 ± 0.026 | Meters |
| Catwalks | This study | str | Cat | — | — |
| Substrate | This study | str | Cat | — | — |
| Fencing | This study | str | Num | 3,371.544 ± 216.587 | Meters |
| Proportion of natural cover | Yamashita, Perotto-Baldivieso [48] | env | Num | 0.786 ± 0.171 | None |
| Proportion of water | Yamashita, Perotto-Baldivieso [48] | env | Num | 0.024 ± 0.003 | None |
| Proportion of woody | Yamashita, Perotto-Baldivieso [48] | env | Num | 0.171 ± 0.074 | None |
| Total monthly precipitation | Menne, Durre [49] | env | Num | 48.955 ± 50.398 | mm |
| Speed limit | This study | ant | Num | 98.389 ± 6.873 | Km/h |
| Proportion of building area | Yamashita, Wester [50] | ant | Num | 0.002 ± 0.007 | None |
| Vehicle traffic | Texas Department of Transportation [38] | ant | Num | 3,398.988 ± 2,819.024 | Vehicles/day |
| Human activity | This study | ant | Num | 6.467 ± 10.417 | Detections |
| Domestic animal activity | This study | ant | Num | 6.818 ± 20.080 | Detections |
| Livestock activity | This study | ant | Num | 1.305 ± 8.352 | Detections |

[1]The data source.

We considered four structural variables for inclusion in the models: openness ratio, presence of catwalks, substrate, and fencing length (S3 Appendix). Openness ratio was calculated as the (height × width) ÷ length (m) and served as a proxy for WCS size [4]. Catwalk presence was a binary variable representing the existence of a catwalk. Two types of catwalks existed (concrete step and dirt bench), but only one site had a dirt bench (S1 Appendix) so these were combined for this analysis. Substrate was categorical with three classes: water, concrete, or dirt with water representing the reference level for analyses. Fencing length was the length of fencing (m) around a WCS. Some WCSs had different fence lengths on either side of the highway so these were averaged for those sites. Other sites shared fencing among multiple WCSs so each of these WCSs received the same fencing length (S1 Appendix).

Four environmental variables were considered in the models: proportion of natural landcover, water, and woody cover within 1 km of the highway and the total monthly precipitation. We chose 1 km for the buffer size because the road effect zone was estimated at 1-km for most mammals in this region [4,56,57]. We calculated the proportions of natural, water, and woody landcover using a classified image of the study area from 2016 National Agricultural Imagery Program (1-m resolution; NAIP) available from the Texas Natural Resources Information Systems. Image classification details can be found in Yamashita, Perotto-Baldivieso [48]. Generally, six classes were identified using a random forest classification model: bare, herbaceous, water, woody, agriculture, and developed. Natural landcover is the sum of bare, herbaceous, water, and woody while water and woody are the proportions of their respective individual land covers. Total monthly precipitation (mm) was obtained from the National Oceanic Atmospheric Administration (NOAA) weather station at the Port Isabel/Cameron County Airport [49,58].

We considered six anthropogenic variables for inclusion in the models: highway speed limit, vehicle traffic, proportion of buildings within 1000 m of the highway, and human, livestock, and domestic animal activity at the WCS. Highway speed limit represented the speed limit at the WCS location and was a proxy for the actual vehicle speed. Vehicle traffic was the AADT at the nearest station to a WCS on the study highway and was obtained from a publicly available TxDOT database

[38]. The proportion of buildings within 1-km of the WCS was calculated from a buildings dataset created from a LiDAR point cloud and classified using software-based tools (see Yamashita, Wester [50] for details). We determined human, livestock, and domestic animal activity from camera trap photographs by calculating the number of independent events for each group in a month. Livestock included cattle, goat, and sheep whereas domestic animals included cats (*Felis catus)* and dogs (*Canis familiaris*).

## Model development

Before analysis, the total detections, successful crossings, and failed crossings datasets were square-root transformed to reduce the influence of hyper-abundant species, such as opossum and raccoon, on the dissimilarity matrix [59]. We calculated a zero-adjusted Bray-Curtis dissimilarity matrix for each response variable [23,52, Ch. 7]. An adjustment of 0.03 was used to represent the smallest possible unit in our data matrix of one detection in a 31 day month [60].

We examined potential predictor variables for outliers, skewness, and multicollinearity, with outliers and skewness examined visually using histograms and scatterplots. Openness ratio, fencing length, human activity, livestock activity, domestic animal activity, and building proportion were left skewed so we $\log(X+1)$ transformed these to reduce the influence of outliers [61]. Multicollinearity was assessed by examining correlation coefficients among predictors and by calculating the variance inflation factor (VIF [52, Ch. 10, 62, Ch. 10, 63, Ch. 12]). While VIF is a subjective tool for assessing multicollinearity [62] and depends heavily on the presence of dummy variables (i.e., categorical variables), VIF provides useful information about potential multicollinearity in a model [64]. Initially, multicollinearity was only assessed within a particular set of predictors (structural, environmental, and anthropogenic characteristics) and no variables were dropped due to high multicollinearity.

To determine which dbMEM axes of spatial and temporal autocorrelation should be included, a forward selection approach based on adjusted Akaike Information Criterion (AICc) was used [53]. To determine the set of structural, environmental, and anthropogenic variables, we separately assessed each set using a step-wise model selection approach with spatial and temporal variables included into the models to isolate the relative effect of each set of predictors. Once the full suite of predictors was selected, we re-assessed multicollinearity using visual examination and VIF scores. We used a liberal threshold for high VIF (VIF > 20) because our models included dummy variables and we were interested in overall model fit, not assessment of individual predictors [62]. Fencing and vehicle traffic both had high multicollinearity so we tested models that dropped each variable for stability and model fit using predicted residual error sums of squares (PRESS) statistics and compared this to the error sums of squares (SSE [63]). Models that dropped fencing had smaller differences between SSE and PRESS, indicating better model fit. Therefore, our final models excluded fencing.

## Variation partitioning

Once we determined the full suite of predictors, we assessed the relative proportion of variation explained by each set of predictors using a variation partitioning approach [52,65,66]. The significance of each set of predictors given the presence of the other sets was assessed using permutation-based tests and distance-based redundancy analysis (dbRDA [61,67]). We calculated the adjusted $R^2$ values using dbRDA for each possible combination of models using Primer v7 (Primer-E, Albany, Auckland, New Zealand) and identified the proportion of variation explained using subtraction of the adjusted $R^2$ values between appropriate models [65]. This process sometimes results in negative variation explained which represents instances where the interaction is interfering with the variation explained by the rest of the model [68]. We completed this analysis manually because existing software cannot accommodate five sets of predictors (the code and associated data are available in S4 Appendix).

## Development and assessment of the predictive model

To model and predict the species assemblage at WCSs, we wanted a model that was not restricted to our current dataset. A dbMEM is data-specific and inclusion of new sites or months would alter the correlation structure, invalidating the

original dbMEM axes used in model development. Therefore, we modelled spatial and temporal autocorrelation using the raw coordinates and month rather than their associated autocorrelation functions [69]. Whereas raw coordinates rarely adequately account for autocorrelation [51], they are an easy-to-implement predictor in a regression model and can partially alleviate the issue of autocorrelation [70]. Additionally, a test of model fit between detrended and non-detrended full models indicated few differences in model fit and prediction power.

A dbRDA is a multivariate regression on the PCO axes [71], so normal regression techniques for prediction and validation of the model can be used. Additionally, despite modelling PCO axes and not actual species detections, tri-plots based on dbRDA can be used to plot the correlation between species detections and PCO scores allowing one to determine the expected species assemblage at a new WCS. Using the relative positions of sites and species in ordination space, one can predict the species assemblage at a particular WCS [52].

We first conducted a PCO analysis on the response variable dissimilarity matrices. We then identified non-trivial PCO axes (i.e., which axes should be interpreted) using several techniques: (1) visual examination of the PCO bi-plots, (2) the broken-stick model, (3) bootstrap resampling of the species data, (4) permutating the values in each column vector, and (5) using the permutation method of (4) but calculating the percent variation as a fraction of the variation remaining after previous axes have been fitted (conditional approach; S5 Appendix [71]). The predictor variables were then centered around their means and ordinary least squares regression was used to calculated fitted PCO scores of the non-trivial PCO axes. We assessed the assumptions of the permutation tests used for significance testing (independent and identically distributed errors) using several techniques [71,72]. We calculated an autocorrelation function and Mantel correlogram to assess independence and plotted the fitted PCOs versus the residuals to test the distribution of the errors [71].

To assess model validity, ideally we would have a sufficiently large training and testing dataset [62,63]. Unfortunately, our system only included 18 WCSs which was not large enough to split into a training and testing dataset, so we used a drop-one-site approach to model validation [62]. Similar to the leave-one-out cross-validation (LOOCV) technique, we excluded one WCS, reran the model, then predicted the excluded WCS using the fitted model. We dropped all months associated with a WCS, because our unit of observation for this study was the WCS not one month at one WCS. To assess model validity, we computed the Mantel correlation between: (1) the full original data and the fitted data of the original model to represent the model fit of the overall model, (2) the data from the drop model and the fitted data from that model to represent the model fit of the drop model, (3) the original data of the dropped site with the predicted values from the drop model to represent the fit between the original data and their predicted values, (4) the fitted data from the overall model for the dropped site and the predicted values of the dropped site from the drop model to represent the difference in fit between the overall model and the drop model, and (5) the full original data and the fitted data and the predicted site from the drop model to represent model fit between the original data and the fitted/predicted values from the drop model. We also plotted the predicted values on the original ordination diagrams to visually assess the accuracy of the prediction.

## Results

### Camera data results

Across all WCSs, the number of camera trap nights varied from 526 to 982. Sites on FM 106 had 550 trap nights (Table 1). On FM 1847, WCS1 and WCS2 had 596 trap nights and WCS3, WCS4, and WCS5 had 546 trap nights. On SH 100, WCS1 and WCS2 had 587 trap nights, WCS3 and WCS4 had 526 trap nights, and WCS3A had 982 trap nights. We detected 22 unique mammal species across all months and time periods, which represented all known medium-large terrestrial mammals in the study area [73]. Body size of species detected was highly variable, ranging from the long-tailed weasel (*Mustela frenata*, 100g [73]) to non-native nilgai (*Boselaphus tragocamelus*, 300 kg [73]). On SH 100, no species were detected in December 2017 at WCS3 and WCS4, the first month of monitoring for both WCSs.

## Model building results

We included 10 axes of spatial autocorrelation in the total detections model, nine axes in the successful crossings model, and seven axes in the failed crossings model. For temporal autocorrelation, we included three axes in the total detections model, five axes in the successful crossings model, and four axes in the failed crossings models. All structural, environmental, and anthropogenic characteristics were included in the successful crossings model but domestic animal and livestock activity were excluded from the total detections and failed detections model. For full description of the model building results, see S3 Appendix.

## Variation partitioning

All sets of predictors (spatial, temporal, structural, environmental, and anthropogenic) explained a significant proportion of the variation in total detections, successful crossings, and failed crossings (all $p < 0.001$). Spatial autocorrelation explained 9.61%, 9.65%, and 6.21% of the variation in total detections, successful crossings, and failed crossings, respectively, while temporal autocorrelation accounted for 1.57%, 4.17%, and 1.64% of the variation, respectively (S4 Appendix).

For total detections, anthropogenic characteristics explained the most variation (2.48%), followed by environmental characteristics (1.41%) and structural characteristics (1.30%; Fig 2). Structural characteristics explained the most variation in successful crossings (4.31%), followed by environmental (3.88%) and anthropogenic characteristics (3.17%; Fig 2). Anthropogenic characteristics explained the most variation in failed crossings (3.97%) followed by structural (3.54%) and environmental characteristics (2.40%; Fig 2). Generally, two-way interactions between structural, environmental, and anthropogenic characteristics explained a negative amount of variation while the three-way interaction between them explained positive amounts of variation (Fig 2). The residual variation not explained by any of the sets of predictors was 51.77%, 55.63%, and 66.60%, respectively. Full contributions to variation partitioning are provided in S4 Appendix.

## Predictive modelling

By not detrending the spatial and temporal autocorrelation, we saw a slight drop in model fit and in Mantel correlation, but this difference did not affect our ability to interpret the models (S5 Appendix). We identified three non-trivial PCO axes for total detections and successful crossings and two non-trivial axes for failed crossings so we limited our analyses to these axes. These axes had $R^2$ values of between 0.40 and 0.67 for the total detections model, 0.38 to 0.54 for the successful crossings model, and 0.33 to 0.45 for the failed crossings model. Our models generally met the assumptions of the permutation tests (S5 Appendix).

When we iteratively dropped one WCS then predicted the site using the fitted model, the total detections model was generally well predicted (Fig 3), while the successful and failed crossings model had mixed results (Figs 4 and 5). Only one site was poorly predicted in the total detections model. Six sites were poorly predicted in the successful crossings model and four sites were poorly predicted by the failed crossings model. From examination of the ordination diagrams, opossums were most associated with WCSs with more human activity while raccoons, nutria, and beaver were associated with WCSs with water substrate (Fig 3D, 4D, and 5D). The majority of other species, including ocelots, grouped with the remaining sites. For full results, including the Mantel correlations, see S5 Appendix. Ordination plots including PCO axis 3 are provided in S6 Appendix.

## Discussion

This study represents the first effort to our knowledge to predict the mammal community composition at a WCS, a critical component allowing transportation managers and biologists to design improved WCSs. Not only is it important to account for spatial and temporal autocorrelation when assessing the relative effects of WCS characteristics, but structural,

A

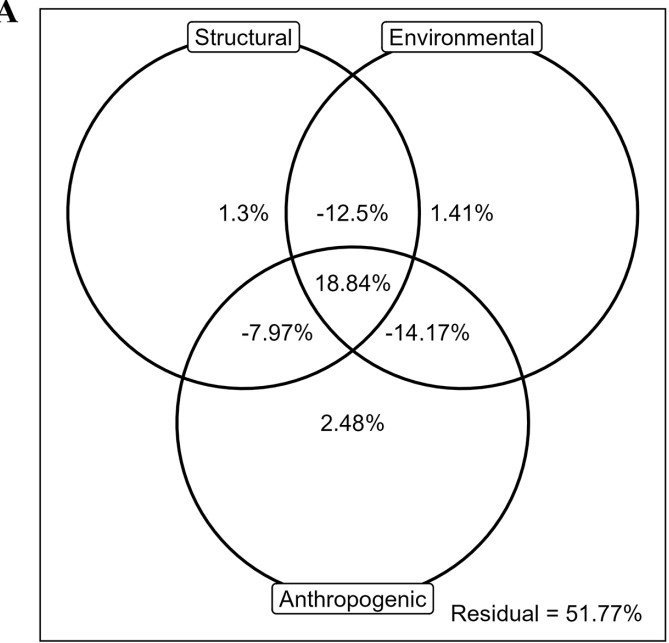

B

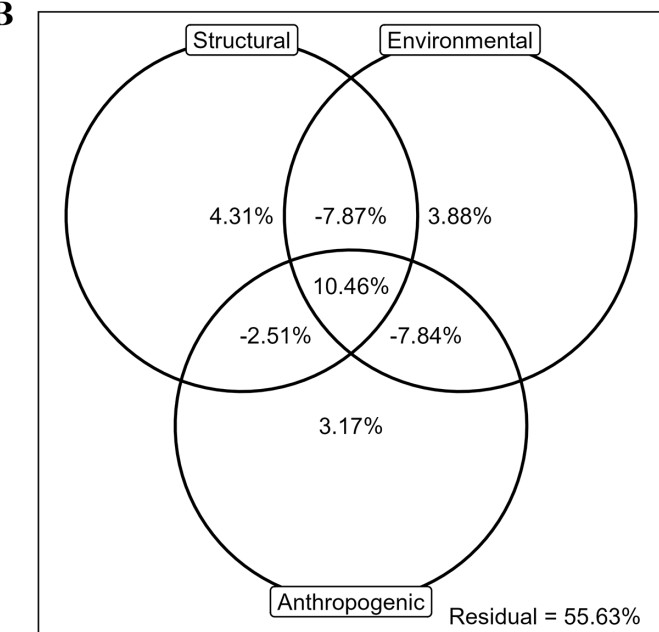

C

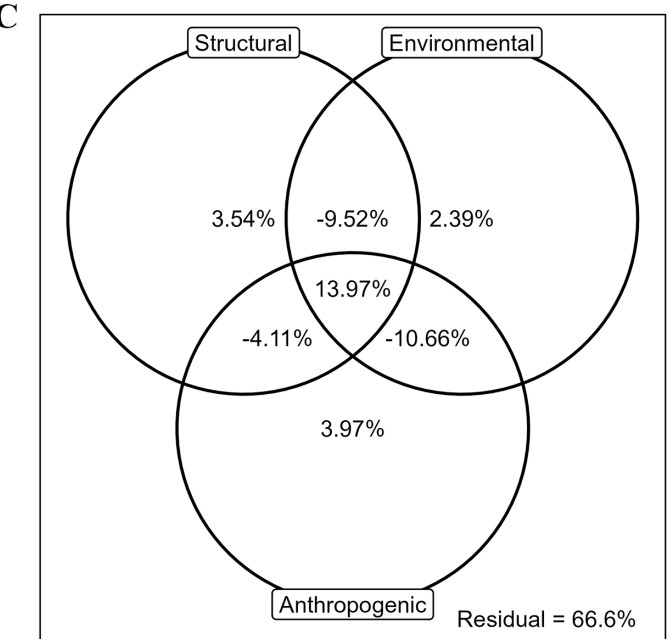

**Fig 2. Variation Partitioning Results: Venn diagram of variation partitioning results showing the percent of variation explained by structural, environmental, and anthropogenic variation and their interactions for total detections (A), successful crossings (B), and failed crossings (C).** Spatial and temporal contributions to the total variation explained and their associated interactions are available in S4 Appendix.

environmental, and anthropogenic characteristics all contributed a significant proportion to the variation in community composition during the first year after construction of WCSs in South Texas. Our models performed relatively well at predicting WCS community composition, particularly for total detections. While model fits were relatively low using the

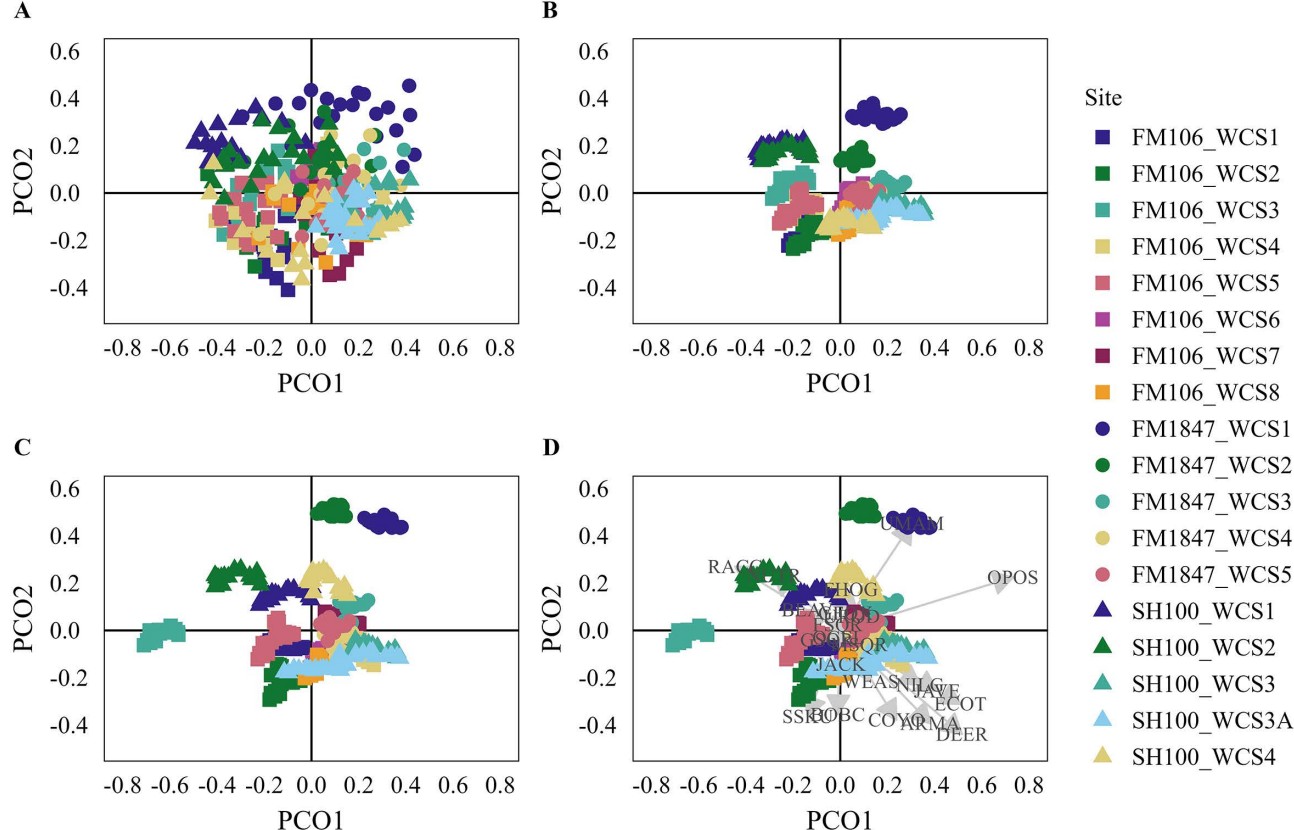

**Fig 3. Ordination Results for Total Detections: Ordination diagrams showing (A) the raw principal coordinates scores (PCO), (B) original fitted PCO scores (these can also be thought of as the output of distance-based redundancy analysis), (C) the predicted PCO scores from the models in which that site was dropped for PCO axes 1 and 2, and (D) the species scores overlayed on the predicted PCO scores for the total detections analysis.** Full species names are provided in S2 Appendix.

drop-one-site method, most WCSs were near enough to their original location in ordination space to reasonably predict the species assemblage.

While structural, environmental, and anthropogenic characteristics did not explain a large proportion of the variation in WCS use, this was significant indicating that these characteristics likely play an important role in determining the effectiveness of a WCS. Generally, structural characteristics explained more variation than environmental characteristics indicating that the structure of the WCS itself may be more important than the surrounding landcover or weather conditions. This conclusion agrees with previous research that has shown that WCS structural characteristics are important during the first-year of post-construction as animals are learning about the WCS and adjusting their movement patterns and behaviors towards using them [16]. However, over time, environmental characteristics will likely become more important as WCSs become more established in the system and animals recognize their usefulness [17].

Interestingly, despite the small $R^2$ values, anthropogenic variables contributed a greater amount of variation than structural characteristics for total detections and failed crossings, and slightly less variation than structural characteristics for successful crossings indicating that anthropogenic characteristics likely play an important role not only in determining whether an animal will be present at a WCS but also whether the animal will successfully cross. Anthropogenic influences on WCS effectiveness have not yet been extensively explored (but see Clevenger and Waltho [2000]), but our results indicate that anthropogenic characteristics may play crucial roles in determining how animals use WCSs. Therefore, more

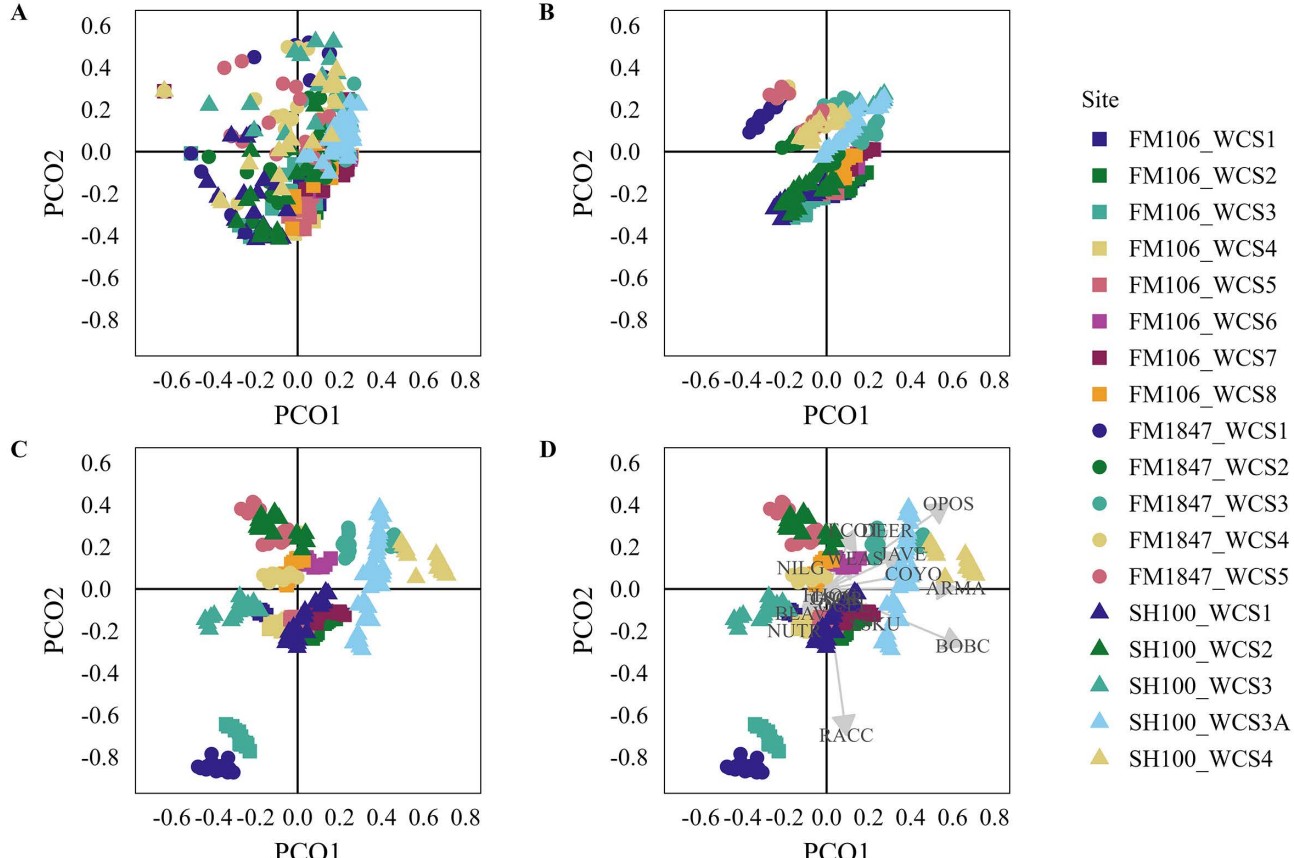

**Fig 4. Ordination Results for Successful Crossings: Ordination diagrams showing (A) the raw principal coordinates scores (PCO), (B) original fitted PCO scores (these can also be thought of as the output of distance-based redundancy analysis), (C) the predicted PCO scores from the models in which that site was dropped for PCO axes 1 and 2, and (D) the species scores overlayed on the predicted PCO scores for the successful crossings analysis.** Full species names are provided in S2 Appendix.

research is critical to evaluate how the human environment impacts WCS use and how future WCS designs can mitigate these effects. This need is especially relevant as more WCSs are designed as multi-use structures for both humans and wildlife [4, Ch. 22]. Because human activity can have strong impacts on wildlife [74,75], it may be necessary to limit human use of multi-use WCSs to allow sensitive and rare species to utilize the WCSs [4, Ch. 76]. Incorporating spatial and temporal autocorrelation into the analysis was clearly important as spatial and temporal variability explained more of the variation in the response variables than structural, environmental, or anthropogenic characteristics. Interestingly, while detrending the correlation using dbMEMs was valuable and contributed to the variation explained, using simplistic representations of space and time also reduced the effects of autocorrelation and provided a useful model for prediction of WCS mammal community composition.

While our models predicted which species will be at a WCS but not necessarily whether they will cross, they still provide valuable insights into the functionality of future WCSs. By examining ordination diagrams, we can identify the potential species assemblage of a new WCS, which is likely the most valuable output from these models. Relative distance between site scores is a proxy for similarities in species assemblages between WCSs so sites that are nearer to each other in ordination space are more likely to have similar species assemblages. By overlaying the correlations between

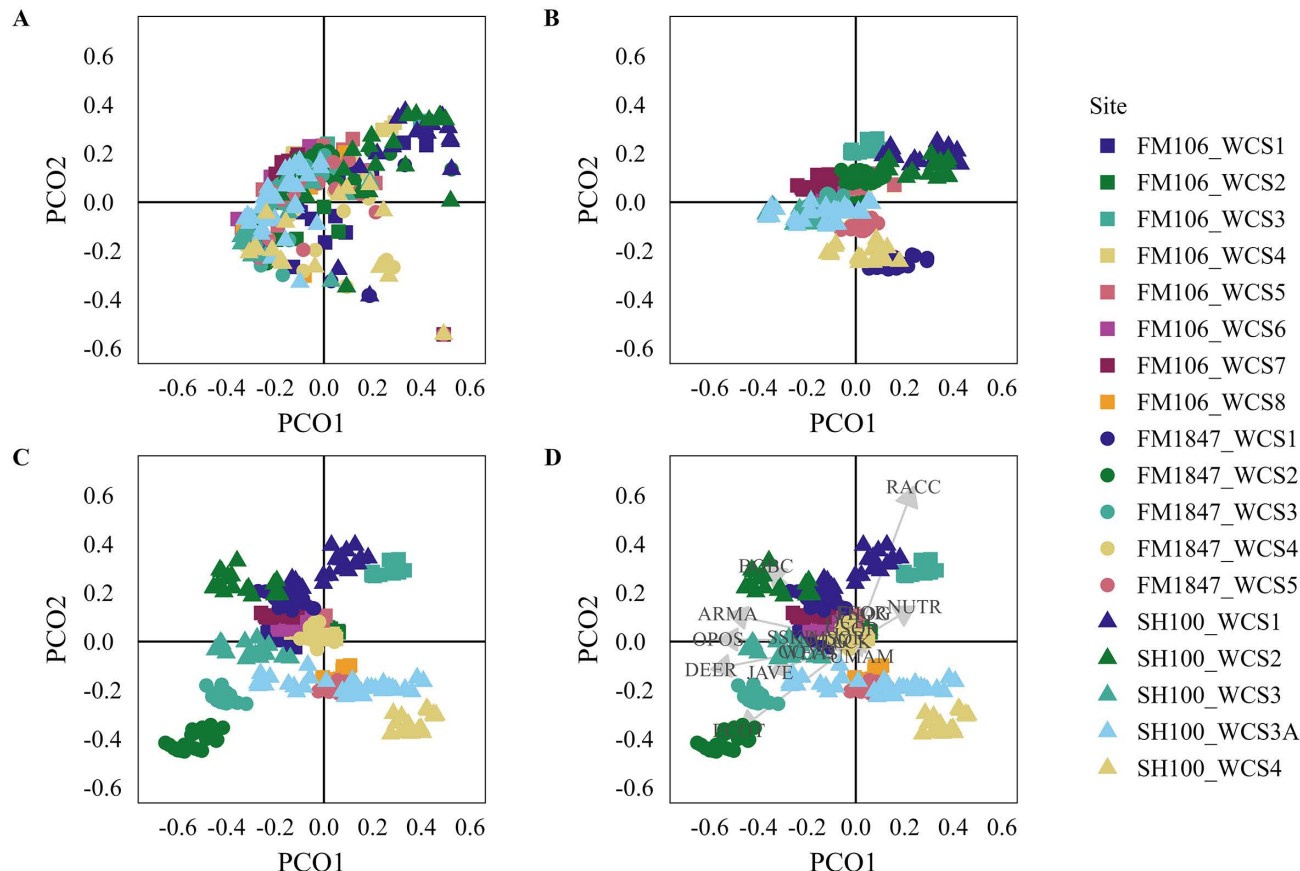

**Fig 5. Ordination Results for Failed Crossings: Ordination diagrams showing (A) the raw principal coordinates scores (PCO), (B) original fitted PCO scores (these can also be thought of as the output of distance-based redundancy analysis), (C) the predicted PCO scores from the models in which that site was dropped for PCO axes 1 and 2, and (D) the species scores overlayed on the predicted PCO scores for the failed crossings analysis.** Full species names are provided in S2 Appendix.

species detections, predictors, and site scores, we can assess individual species' preferences for particular characteristics at each WCS (S6 Appendix).

Our model performed relatively well for total detections but performed much worse for successful crossings and failed crossings. The presence of an animal at a WCS as represented by the total detections at a WCS is likely dependent on the characteristics of the WCS itself so the selected predictors probably provide a good estimate of presence at a WCS. However, the crossing rate at a WCS depends on more dynamic factors like the presence of vehicles at the time of use, noise or light conditions, or behavioral considerations at the time [77] that are difficult or impossible to measure with standard camera traps. For example, opossums in South Texas are less likely to cross through a WCS when there was more vehicle noise at the time the opossum was present [78]. Additionally, the WCS characteristics measured in this study may not be important for some species. For example, WCSs in South Texas are intended for ocelots so most are too small for white-tailed deer (*Odocoileus virginianus*) or nilgai. These species may show up at a WCS because of food or water availability and therefore were included in the community-level analyses but are unlikely to attempt to use a WCS because it is too cramped. These species were included in the analyses because some WCSs were large enough for them and both did successfully cross in these locations. Interestingly, smaller WCSs had the highest species diversity and ocelots were only detected at smaller WCSs. For those species that physically can use a WCS, the decision to cross is likely

dependent on not only the WCS characteristics, but behavioral decisions and finer-scale cues such as vehicle noise. Additional covariates representing more dynamic factors (namely noise and light) at WCSs are likely necessary to effectively model successful and failed crossings at WCSs.

We have successfully provided a useful tool for predicting WCS use in other regions. Despite our models not being perfect, they provided a powerful framework for assessing both WCS effectiveness and potential use at new, unbuilt WCSs. However, the model, especially selection of appropriate predictors, should be tailored to the study area. Important predictors may vary so it is important to select relevant predictors for the study system. This selection of predictors can be aided by the literature or expert opinion, as well as model selection procedures.

Our models make two important assumptions about the species assemblage. First, our models assume that the species assemblage around the existing and new WCSs are the same. Our models do not predict species that were never detected and therefore should only be applied to areas that have similar species composition to the modelled sites. Otherwise, the predicted species assemblage will not reflect the potential species assemblage at the new site. Therefore, it is important to understand the diversity of the system under study and only include WCSs in the model that likely have similar community compositions to the new WCS. It may be possible to truncate species assemblage information to those species that are shared between existing and future WCSs. While this option may provide opportunity to expand the model to new locations, this was not tested, and warrants future research. Second, the existing structures should be at least somewhat effective for the target species. Ideally, as was the case in this study, the target species uses some of the WCSs. This increases confidence that the existing WCSs have characteristics that the target species prefers. Additionally, the WCSs with the highest diversity were visited by ocelots, lending credence to our community composition approach to predicting target species WCS use.

It is also important to recognize and consider that prediction models based on regression should not be used to predict outside the range of the predictors [62]. The relationship between the predictors and response variables may not be linear beyond the range of predictors so extrapolating from models should be done with caution [62]. In our models, WCSs that were poorly predicted typically had unusual species assemblages or predictor values that were on the extremes of modelled predictors. Other modelling approaches such as regression trees or neural network models may better account for unobserved variability in data and should be explored to better understand how species assemblages respond to WCS characteristics.

Our study was limited to the first-year after construction of WCSs which likely affects the long-term prediction ability of our models, particularly beyond the first year when environmental characteristics are likely to become more important [17]. Our aims were to examine mammal use of newly constructed WCSs to develop a predictive model of WCS use that can be used by biologists, practitioners, transportation planners, and engineers, to subsequently design and build more effective WCSs. While we recognize that factors affecting WCS use change over time, there is no reason why our model could not be applied to prediction through time instead of new sites. In addition, if models include factors that are likely important over multiple time scales, then they are likely to be valid for prediction through time. Because our aim was to predict entirely new WCSs which we believe are of greater conservation value than predicting existing WCS use into the future, it provides managers with a tool for designing effective new WCSs. Incorporating statistical interactions between time and predictors of interest would allow researchers to examine changes in the relative effects of predictors through time potentially allowing for prediction in space and time. Long-term monitoring is essential to determining effective WCSs and results of long-term modelling can be incorporated into our model to design effective WCSs over the short- and long-term. Models based on shorter time intervals, such as this one, may still provide useful information about WCS use and may allow managers to modify WCSs before they are constructed to increase WCS use by target species.

We provide a framework for managers and biologists to design more effective WCSs using relatively easy-to-collect camera trap data. When WCSs are built, long-term monitoring remains essential to understanding their effectiveness, and as this study clearly shows, this monitoring, particularly of early WCSs, can be used to develop predictive models

to inform future WCS designs and ultimately enable managers to predict the species assemblage at these locations. By utilizing our model, scientists, practitioners, and transportation planners would have the ability to tailor WCS designs to desired species assemblages. Finally, this framework can be applied wherever existing WCSs are located by utilizing relevant predictors and local species assemblages. To design and implement effective WCSs, especially for use by threatened species, requires scientists, land managers, and the public to develop creative solutions to road-related conservation issues.

## Supporting information

**S1 Appendix. Description of the characteristics used to predict the mammal community composition of each wildlife crossing structure (WCS).**
(DOCX)

**S2 Appendix. Description of species detected on camera.**
(DOCX)

**S3 Appendix. Detailed results of the model building process showing the model selection process for the spatial, temporal, structural, environmental, and anthropogenic characteristics of wildlife crossing structures (WCSs).**
(DOCX)

**S4 Appendix. Variation partitioning model construction and results.**
(DOCX)

**S5 Appendix. Model selection and validation results for the full and predictive models.**
(DOCX)

**S6 Appendix. Ordination diagrams from the distance-based redundancy analysis results.**
(DOCX)

## Acknowledgments

We thank the Texas Department of Transportation for providing access to wildlife crossings. We appreciate the many graduate and undergraduate students and research technicians who provided essential help with collection of camera data and processing of photographs. We thank E. Brookover, P. Glover-Kapfer, M. Cherry, S. Riley, D. Wester, and two anonymous reviewers for their many helpful comments that improved the manuscript. This is manuscript #24–136 of the Caesar Kleberg Wildlife Research Institute.

## Author contributions

**Conceptualization:** Thomas J. Yamashita, John H. Young Jr., Jason V. Lombardi.

**Data curation:** Thomas J. Yamashita, Kevin W. Ryer.

**Formal analysis:** Thomas J. Yamashita.

**Funding acquisition:** Daniel G. Scognamillo, Richard J. Kline, Michael E. Tewes, John H. Young Jr., Jason V. Lombardi.

**Investigation:** Thomas J. Yamashita, Daniel G. Scognamillo, Kevin W. Ryer, Jason V. Lombardi.

**Methodology:** Thomas J. Yamashita, Jason V. Lombardi.

**Project administration:** Daniel G. Scognamillo, Kevin W. Ryer, Richard J. Kline, Michael E. Tewes, John H. Young Jr., Jason V. Lombardi.

**Resources:** Richard J. Kline, John H. Young Jr..

**Software:** Thomas J. Yamashita.

**Supervision:** Michael E. Tewes, Jason V. Lombardi.

**Validation:** Thomas J. Yamashita.

**Visualization:** Thomas J. Yamashita.

**Writing – original draft:** Thomas J. Yamashita.

**Writing – review & editing:** Thomas J. Yamashita, Daniel G. Scognamillo, Kevin W. Ryer, Richard J. Kline, Michael E. Tewes, John H. Young Jr., Jason V. Lombardi.

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
