## [Decision Letter · Decision Letter 0]

26 Jun 2025

Dear Dr. Yamashita,

Thank you for submitting your manuscript to PLOS ONE. After careful consideration, we feel that it has merit but does not fully meet PLOS ONE’s publication criteria as it currently stands. Therefore, we invite you to submit a revised version of the manuscript that addresses the points raised during the review process.

Dear authors, considering that the reviewers issued a favorable opinion, with some suggestions. Please ask them to make the adjustments and return the paper (with the adjustments highlighted) so that we can continue the process. Kind regards.

We look forward to receiving your revised manuscript.

Kind regards,

Julio Cesar de Souza, Ph.D.

Academic Editor

PLOS ONE

Journal Requirements:

This work was funded by the Texas Department of Transportation.

We thank the Texas Department of Transportation for providing project funding and access to wildlife crossings. We appreciate the many graduate and undergraduate students and research technicians who provided essential help with collection of camera data and processing of photographs. We thank E. Brookover, P. Glover-Kapfer, M. Cherry, S. Riley, D. Wester, and two anonymous reviewers for their many helpful comments that improved the manuscript. This is manuscript #24-136 of the Caesar Kleberg Wildlife Research Institute.

This work was funded by the Texas Department of Transportation.

5. Thank you for uploading your study's underlying data set. Unfortunately, the repository you have noted in your Data Availability statement does not qualify as an acceptable data repository according to PLOS's standards.

6. In the online submission form, you indicated that code and other data used in this manuscript are available on GitHub at https://github.com/tomyamashita/WCSeffectiveness. Due to the the presence of endangered species in camera trap data, camera trap data is available upon request. The functions used for processing camera data are available in the cameraTrapping package on GitHub at https://github.com/tomyamashita/cameraTrapping .

7. Please amend the manuscript submission data (via Edit Submission) to include author John H. Young Jr.

8. Please amend your authorship list in your manuscript file to include author John H. Young.

9. We note that Figure 1 in your submission contain map/satellite images which may be copyrighted. All PLOS content is published under the Creative Commons Attribution License (CC BY 4.0), which means that the manuscript, images, and Supporting Information files will be freely available online, and any third party is permitted to access, download, copy, distribute, and use these materials in any way, even commercially, with proper attribution. For these reasons, we cannot publish previously copyrighted maps or satellite images created using proprietary data, such as Google software (Google Maps, Street View, and Earth). For more information, see our copyright guidelines: http://journals.plos.org/plosone/s/licenses-and-copyright.

Additional Editor Comments:

Dear authors, considering that the reviewers issued a favorable opinion, with some suggestions.

Please ask them to make the adjustments and return the paper (with the adjustments highlighted) so that we can continue the process.

Kind regards.

Reviewers' comments:

Reviewer's Responses to Questions

**Comments to the Author**

1. Is the manuscript technically sound, and do the data support the conclusions?

Reviewer #1: Partly

Reviewer #2: Yes

2. Has the statistical analysis been performed appropriately and rigorously?

Reviewer #1: I Don't Know

Reviewer #2: Yes

3. Have the authors made all data underlying the findings in their manuscript fully available?

Reviewer #1: No

Reviewer #2: Yes

4. Is the manuscript presented in an intelligible fashion and written in standard English?

Reviewer #1: Yes

Reviewer #2: Yes

Reviewer #1: Article is a bit not clear when you do not explain body size of mammals and their adaptive ecology. I suggest you re-write with addition of body size and nature of structures? I wonder have you mentioned about bio-engineering structures. It may some time attract or repel movement? Also elaborating drop-onsite method clearly would be good for readers.

Reviewer #2: 1.     Line 35. Add a comma before "so" to properly separate the independent clauses.

2. The structure of Table 1 is excessively complex. There is too much information in this table. This table could be further broken down into two or more; e.g., information such as “timeline for WCS construction” could be a distinct table.

3. Generally, predictive modeling should include both a training dataset and a testing dataset. This study did not adhere to this rule. Was a technique such as cross-validation, transfer learning, and leveraging a pretrained model used? A model without a separate test set could be difficult to assess for its true generalization ability and be misleading.

4. Why do you not detrend the spatial and temporal autocorrelation in this study?

5.     Line 503 “Our model performed relatively well for total detections but performed much worse for successful crossings and failed crossings.” A further explanation is needed regarding the poorer performance in both successful and failed crossings. This is a limitation that needs to be worked upon in the future study.

**Do you want your identity to be public for this peer review?** For information about this choice, including consent withdrawal, please see our Privacy Policy

Reviewer #1: No

Reviewer #2: No

---

## [Author Response · Author response to Decision Letter 1]

24 Sep 2025

Journal and Editor Comments

We have made all editorial changes required by the journal and updated our funding, acknowledgements, and data availability statements. We have moved our data repository to FigShare and provided DOIs for it and the R package used for data processing. Figure 1 used imagery from the national agriculture imagery program and is in the public domain. The authorship issue was due to a missing “Jr.” in John Young’s name in the submission portal.

Reviewer #1:

Article is a bit not clear when you do not explain body size of mammals and their adaptive ecology. I suggest you re-write with addition of body size and nature of structures? I wonder have you mentioned about bio-engineering structures. It may some time attract or repel movement? Also elaborating drop-onsite method clearly would be good for readers.

We do mention that most of the WCSs were too small for large mammals such as deer or nilgai; however, both species used WCSs when they were large enough for them so we included them in the analyses. We believe that generally body size is not relevant for the purposes of this study where we were interested in the community composition of wildlife crossings, not in examining allometric relationships with crossing use. While this could be an interesting future avenue of study, in our study system, not all crossings in our system are large enough for all species, so it would be difficult to examine body size effects on crossing rates. As such, we have added the range of body size to our description of the species captured and added some additional discussion of deer and nilgai WCS use.

We are unclear what the reviewer means about bio-engineered wildlife crossings. While WCSs are often designed for particular species, this is often done based on assumptions about their behavior and preferences. Our study could help improve future WCS design by providing a method to test the potential efficacy of a WCS for a suite of species of interest and we explicitly state this in our discussion.

We have clarified that our drop-one-site method is similar to the leave-one-out-cross-validation technique for assessing model validity. In our case, the unit of observation is the WCS itself so our unit of observation for cross validation was all months in a WCS rather than a single sample (one month in one WCS). See below for further explanation of the method.

Reviewer #2:

1. Line 35. Add a comma before "so" to properly separate the independent clauses.

We have added the requested comma.

2. The structure of Table 1 is excessively complex. There is too much information in this table. This table could be further broken down into two or more; e.g., information such as “timeline for WCS construction” could be a distinct table.

We have split Table 1 into two tables, one for the characteristics of the WCSs on the study roads and the other for the timeline of construction and monitoring of the WCSs.

3. Generally, predictive modeling should include both a training dataset and a testing dataset. This study did not adhere to this rule. Was a technique such as cross-validation, transfer learning, and leveraging a pretrained model used? A model without a separate test set could be difficult to assess for its true generalization ability and be misleading.

We agree that for predictive modeling, ideally we should have both a training dataset and a testing dataset and we acknowledge this in the methods, line 359. Because our sample size of WCSs was small (n = 18) and highly variable, we opted against leaving out a set of WCSs as a testing dataset. Instead we used a technique similar to LOOCV where we excluded all months from one WCS, reran the model, then predicted the dropped WCS using the fitted model. We originally described this in the paragraph starting on Line 363 and have clarified how our drop-one-site approach is similar to LOOCV.

4. Why do you not detrend the spatial and temporal autocorrelation in this study?

We detrended the spatial and temporal autocorrelation using a dbMEM when we ran our variation partitioning. However, for our predictive modeling, we wanted to create a model that could be used to predict future use of other (unbuilt) wildlife crossings in the system and be generalizable to other systems. The issue with using dbMEM to detrend autocorrelation is that the dbMEM axes that get included are model specific and you cannot add new values of dbMEM axes for additional testing data without changing the whole model. This makes the model useless for prediction purposes. We did, however, test the differences in model fit between a model fitted with raw coordinates and months and a detrended model and found little to no difference in the model fits, indicating that, for our purposes at least, detrending the autocorrelation was likely not necessary and results in a less useful predictive model. We have added some clarification to Lines 334-340.

5. Line 503 “Our model performed relatively well for total detections but performed much worse for successful crossings and failed crossings.” A further explanation is needed regarding the poorer performance in both successful and failed crossings. This is a limitation that needs to be worked upon in the future study.

We believe that the poor performance of the successful crossings and failed crossings models was due to the additional factors that influence actual crossing rates, such as the presence of vehicles, noise or light levels, or the individual’s behavior/reason for being at the WCS itself. Another study in the same study area showed that opossums avoid crossing through WCSs when there is more vehicle noise [1]. Additionally, based on observation and videos in the study system, many species forage around WCSs and they may approach the WCS, appearing to interact with the structure on still image camera traps, as used in this study, but not actually have intentions of crossing.

References

1. Yamashita TJ, Tanner AM, Tanner EP, Scognamillo DG, Tewes ME, Young Jr JH, et al. The importance of soundscapes in monitoring wildlife crossing structures. Ecological Applications. In Review.

---

## [Editor Report · Decision Letter 1]

7 Oct 2025

Predicting species assemblages at wildlife crossing structures using multivariate regression of principal coordinates

PONE-D-24-56498R1

Dear Dr. Yamashita,

We’re pleased to inform you that your manuscript has been judged scientifically suitable for publication and will be formally accepted for publication once it meets all outstanding technical requirements.

Kind regards,

Julio Cesar de Souza, Ph.D.

Academic Editor

PLOS ONE

Additional Editor Comments (optional):

Considering that the reviewers considered a Minor Revision and that the authors accepted the suggestions and made changes to the text,

I am in favor of publishing this paper.

Best regards

Julio Souza
---

## [Editor Report · Acceptance letter]

PONE-D-24-56498R1

PLOS ONE

Dear Dr. Yamashita,

I'm pleased to inform you that your manuscript has been deemed suitable for publication in PLOS ONE. Congratulations! Your manuscript is now being handed over to our production team.

Kind regards,

on behalf of

Dr. Julio Cesar de Souza

Academic Editor

PLOS ONE